# Morphological, Physiological, and Molecular Stomatal Responses in Local Watermelon Landraces as Drought Tolerance Mechanisms

Kelebogile Madumane, Lesego T. Sewelo, Metseyabeng N. Nkane, Utlwang Batlang and Goitseone Malambane * 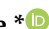

Department of Crop and Soil Sciences, Botswana University of Agriculture and Natural Resources, Private Bag, Gaborone 0027, Botswana; 201700403@buan.ac.bw (K.M.); 201700117@buan.ac.bw (L.T.S.); 201200327@buan.ac.bw (M.N.N.); ubatlang@buan.ac.bw (U.B.)
* Correspondence: gmalambane@buan.ac.bw

**Abstract:** Drought-tolerant plants have become a convenient model to study the mechanisms underlying drought tolerance in order to improve susceptible domesticated relatives. Various studies have shown that local landraces possess superior qualities that help them survive in harsh environmental conditions. One of the key mechanisms that helps with tolerance in crops is timely stomatal regulation. In this study, the physiological, morphological, and molecular stomatal responses in three drought-tolerant landraces (Clm-01–03) and hybrid (Clm-04) watermelons were evaluated under drought stress. The watermelon plants were grown under a water deficit (complete withholding of water) and non-stress conditions. The highest SPAD values were recorded for the Clm-03 and Clm-02 (50 ± 3) watermelon genotypes, and the lowest for Clm-04 (27 ± 0.37), showing this genotype's tolerance and ability to maintain its systems during drought stress. Fluorescence parameters also gave important clues to the tolerant genotypes of Clm-02 and Clm-03 under drought stress, while the domesticated genotype showed a slow response to fluorescence parameters, which could lead to damage to the photosynthesis apparatus. During the drought period, the wild watermelon was found to have a limited stomatal opening as the drought progressed, and on day 9, it had the smallest opening of 23.1 ± 1.2 μm compared to any other genotype; most importantly, upon re-watering, it showed more rapid recovery than any other genotype. This was also expressed by mRNA quantification of stomatal aperture TFs, with an eight-fold increase in *Cla004380* TFs recorded for wild watermelon. All of these mechanisms have been attributed to the tolerance mechanisms of the drought-tolerant watermelon genotype. This study provides important insight into the stomatal responses of probable tolerant watermelon accessions and suggests that improving the stomatal aperture of susceptible domesticated species would also improve their tolerance.

**Keywords:** climate change; drought stress; wild watermelon; stomatal regulation; stomatal aperture





## 1. Introduction

Climate change poses significant challenges to sustainable agriculture, particularly in the context of drought stress. Water deficit, a component of drought, alters the physiological, morphological, and biochemical properties of crops, as well as their molecular characteristics, which impairs their growth and development. Sub-Saharan Africa has been suggested as a location experiencing adverse effects of climate change, with some preliminary signs like an increased frequency of drought observed over the past few decades [1,2]. This is anticipated to have several negative effects on dry-land crop production [3,4]. As a result, the drought stress tolerance of plants is becoming a major focus of research to establish possible ways of adapting.

To survive under water-deficit conditions, plants must maintain their water status and homeostasis. Therefore, plants have evolved various strategies, including drought escape, drought avoidance, and drought tolerance, which enable the prevention of water loss and the survival of water deficit conditions. Stomatal regulation has been found to be

important for drought stress tolerance in plants through drought avoidance, which involves sustaining important physiological processes [5]. Stomatal closure has been documented as one of the first responses to drought stress in plants [5]. Its purpose is to reduce water loss through transpiration, and it is mainly influenced by soil moisture content as compared to leaf water status.

Another morphological alteration due to moisture deficit stress is a decrease in stomata number. Reduced stomatal opening reduces net $CO_2$ uptake, transpiration, and net photosynthesis, mechanisms plants use to survive in variable conditions, including water deficit [6]. Under extreme drought stress, stomata can completely close, and this is highly influenced by the plant species. The wild relatives of some cultivated crops have been shown to be more tolerant to harsh environmental stresses as they have superior morphological, anatomical, and physiological traits [7]. These traits include efficient control of transpiration by regulating stomata, which is important in wild crop relatives to manage the efficiency of $CO_2$ assimilation and transpiration under abiotic stress conditions, such as xerophytic wild barley, which has low stomatal conductance [8]. Additionally, compared to the kidney-type stomatal structure mostly observed in dicotyledons, the stomatal structure in Poaceae crops, with two guard cells and two secondary guard cells, has great potential for improving water utilization in crops and overall drought tolerance [9,10]. Tolerant species regulate the status of their stomata to enable carbon fixation and assimilation while also increasing water use efficiency. Most cultivated crop species have wild relatives that exhibit tolerance to abiotic stresses. Therefore, to develop efficient crop improvement production methods, wild species can be used as a good source for studying crop responses to environmental stresses.

Crop landraces and their wild relatives have been shown to be important in advancing our understanding of tolerance mechanisms, as they have been shown to have naturally acquired these mechanisms, which can be harnessed and used to increase the tolerance of domesticated crop species to both biotic and abiotic stresses [11,12]. It is important to harness the natural patterns of diversity of wild watermelon (*Citrullus lanatus* sp.), which is a species of the Cucurbitaceae family that inhabits the Kalahari Desert in Botswana. This crop survives annually in the desert from spring through to summer. The climate of the Kalahari Desert is very severe for plant growth [13], with little rainfall, which causes an accumulation of salts on the soil surface, strong summer light, and less available water that causes oxidative and heat damage to plants.

The two landraces used in this study are commonly cultivated by dry-land subsistence farmers in Botswana and have been shown to be extremely tolerant to drought. These landraces have always had above-average yields, even in the worst rainy season. Thus, there have been calls for extensive studies on them in order to understand their survival mechanisms and exploit them for crop domestication. The objective of this study was to evaluate the stomatal aperture in local watermelon accessions (Clm-01–03) as a possible tolerance mechanism, which has been well documented as the first response to drought stress. The study involved evaluating the morphological, physiological, and molecular response patterns of traits associated with stomatal aperture in accessions with different drought tolerance levels. We hypothesized that stomatal movement plays a significant role in the tolerance mechanisms of plants and that local landraces have a superior stomatal response to drought stress compared to domesticated hybrid watermelons. By elucidating these important mechanisms, we expect to provide valuable insights into and technical support for investigating stomatal movement as an important element in improving crops for drought tolerance.

## 2. Materials and Methods

### 2.1. Study Site and Plant Materials

This experimental research was carried out in Sebele at the greenhouses of the Botswana University of Agriculture and Natural Resources and was repeated three times. Four watermelon accessions were used in this study (Table 1). Among them, one was a

wild species, two were local landraces, and one was a commercial hybrid. The landraces were sourced from the Botswana National Plant Genetic Resource Center (BNPGRC) seed bank in Sebele, while the hybrid watermelon seeds (crimson sweet) were purchased from an agricultural shop in Gaborone.

**Table 1.** Information on *Citrullus lanatus* (watermelon) accessions used in this study.

| Code | Accession Type | Cultivation | Source | Type of Use | Flesh Color |
|------|----------------|-------------|--------|-------------|-------------|
| Clm-01 | Landrace | Domesticated | BNPGRC | Fresh eating | Red |
| Clm-02 | Landrace | Domesticated | BNPGRC | Cooking | Yellow |
| Clm-03 | Landrace | Wild | BNPGRC | Livestock feed | White |
| Clm-04 | Hybrid | Domesticated | Commercial shop | Fresh eating | Red |

*2.2. Planting and Design of Experiment*

The watermelon seeds were planted in wooden drought stress boxes in the greenhouse. One box was used for each moisture treatment. The seeds were planted in rows, with each row representing a single genotype.

All seeds were adequately irrigated every day for 14 days after planting to promote the development of a root system without water stress. Moisture stress was induced by completely withholding irrigation for 9 days, and re-watering was carried out immediately to collect recovery data. Control plants were continually watered per the pre-treatment schedule to maintain soil moisture to near-field capacity throughout the experiment. Management practices were carried out according to the needs of the crop, including weeding, cultivation, and pest control.

Data were collected at 3-day intervals for a period of 15 days, with day 0 representing the control prior to stress initiation and day 12 representing post-re-watering or recovery.

*2.3. Soil Water Content*

Soil moisture content was measured every data collection day using a moisture probe meter (MPM-160-B) with 12-bit resolution (ICT International Pty Ltd., Armidale, NSW, Australia). A calibrated soil moisture sensor was inserted at a depth of 10 cm, and readings were taken and recorded [14].

*2.4. Measurement of Physiological Parameters*

Stomatal conductance (mmol $H_2O$ $m^{-2}$ $s^{-1}$) was measured using a leaf porometer (Decagon Devices, Inc., Pullman, WA, USA). Measurements were taken from 3 fully grown fresh leaves that were selected from 3 random plants of each watermelon genotype. Measurements were recorded by placing the leaf porometer sensor on the leaf for 60 s.

*Leaf relative water content (RWC) was computed by the following method:* [15]

$$RWC = 100 \, [FW - DM]/[TW - DM].$$

where FW is fresh weight (g), TW is turgid weight, and DM is dry weight.

The leaves were weighed (FW) immediately after detaching from plants, then turgid weight (TW) was determined by soaking the leaves in sterile water overnight at 4 °C, and the leaves were oven-dried at 70 °C until they reached a constant weight, then dry weight (DW) was recorded.

*Chlorophyll fluorescence parameters*

The chlorophyll fluorescence parameters maximum quantum yield of PSII (Fv/Fm), qL, non-photochemical quenching (PhiNPQ), ratio of incoming light lost via nonregulated processes (PhiNO), quantum yield of photosystem II (Phi2), and photosystem 1 (PS1) active centers were measured between 10:00 and 12:00 on a cloudless day. A MultiSpeQ (version 2.0, PhotosynQ Inc., East Lansing, MI, USA) was used to take readings of all fluorescence parameters under ambient greenhouse conditions [16]. Chlorophyll content (SPAD) was also measured using the same instrument and procedure.

*Stomatal morphology and characteristics*

At 3-day intervals post-initiation of stress treatment, 3 leaves per treatment were harvested from randomly selected plants to observe the stomatal morphology. Stomata imaging was conducted using the nail polish method as described in [17]. Leaf imprints were collected by applying nail polish to the adaxial and abaxial surfaces of the leaves of watermelon genotypes before detaching the leaves. Clear nail polish (Insta-Dri Topcoat, Sally Hansen, New York, NY, USA) was applied in a thin layer and left to air dry for 5 min. The dried nail polish was peeled off using clear tape (Crystal Clear Office Tape, Winc, Mascot, NSW, Australia) and mounted on a microscope slide. Stomata on leaf imprints were observed using a Carl Zeiss Axio Observer 5 inverted biological microscope (Zeiss, Jena Oberkochen, Baden-Württemberg, Germany) connected to an AxioCamMR5 digital camera and a 40× objective. Images of leaf stomata imprints and measurements of stomatal apertures were taken using ZEN 2011 (blue edition) v1.0 imaging software.

*2.5. Expression Analysis of Stomatal Aperture Genes and Transcription Factors (TFs)*

Sample collection and total RNA isolation:

Leaf tissues of 4 different watermelon genotypes were harvested for use in qRT-PCR analysis. One random leaf was harvested from each of 3 randomly selected watermelon plants per variety and treatment, with each leaf representing a biological replicate. The leaves were collected in liquid nitrogen and immediately stored at −80 °C until RNA extraction.

Total RNA was isolated from the frozen samples using a Quick-RNA™ Plant Miniprep kit (Zymo Research, Irvine, CA, USA). The integrity of the extracted RNA was verified using 2% agarose gel electrophoresis. The concentration and purity of RNA were measured using a NanoDrop 2000 spectrophotometer (Thermo Fisher Scientific, Waltham, MA, USA.

cDNA synthesis:

DNA synthesis from an RNA template via reverse transcription was carried out using a ProtoScript® II First Strand cDNA Synthesis Kit (New England Biolabs Inc., Ipswich, MA, USA) with oligo DT primers. The cDNA was checked for amplification success using the designed primers and reference primers.

Primer design:

Primer sequences of the target genes and reference genes used for normalization were designed with Primer3Plus (http://primer3plus.com/cgi-bin/dev/primer3plus.cgi, accessed on 16 May 2023) based on the sequences of important genes related to stomatal apertures in plants. First, the gene of interest of a model crop was BLAST searched in the Cucurbit Genomic Database to identify homologs in the watermelon gene sequence. The target sequence was then used to design specific primers using a Primer3Plus online tool with the set parameters.

Targeted genes (Table 2):

**Table 2.** Plant transcriptional regulators with a function in stomatal movement and primer sequences used for amplifying genes responsible for stomatal aperture and reference genes in wild watermelon.

| Name | Pathways | Role | Target TFs of Stomatal Aperture and Reference Genes in Wild Watermelon |
|---|---|---|---|
| *Cla017012* | dark/light, ABA | opening | F_CCAATACTGGGTTGCTTAGATGTAG<br>R_GTTCTTTGTGGAAGGTATGAAGCTA |
| *Cla017389* | ABA | opening | F_TCTCTACACTGTTCCTGAAAATTCC<br>R_TGTTCTGACCAAAACCTAATCTCTC |
| *Cla016849* | ABA | closure | F_ACATCTTCATGCACTAAACAGAGTG<br>R:GATCCATTAGAGATGCTTGTGATCT |
| *Cla004380* | ABA | closure | F_GTGGAAGAAGCTCTACAAAGTCAAC<br>R_AAGAGTGTCTTCTTCCTGGTTGTAA |
| *Glyceraldehyde 3-phosphate dehydrogenase(GAPDH)* | | | F_CTGGCAGTACTTTGCCAACA<br>R_AGGATTGGAGAGGAGGTCGT |
| *Tubulin* | | | F_CAGCACTCCTAGCTTTGGTGA<br>R_CGGGGAAATGGGATTAGATT |

Quantitative-RT PCR analysis:

The analysis was run in a final volume of 20 μL, with 50 ng cDNA samples containing 0.5 ng each of forward and reverse primer, 10 μL qPCR master mix, and 5 μL water. The reactions were performed on a Bio-Rad CFX Connect real-time PCR detection system (Bio-Rad Laboratories, Inc., Hercules, CA, USA) with two technical replicates for each of the three biological replicates. Negative controls were PCR mixtures without cDNA templates. The PCR conditions were set as follows: initial denaturation at 95 °C for 30 s, followed by 45 cycles of denaturation at 95 °C for 5 s, annealing at the melting temperature of the primers for 15 s, and extension at 72 °C for 10 s. Melting curve analysis was performed at the end of amplification to verify the specificity of primers used [18]. The 2-ΔΔCT method was used to estimate relative RNA expression [19].

### 2.6. Statistical Analysis

Data analysis was performed with R v.4.3.1 software (2023). An analysis of variance (ANOVA) of datasets was performed to determine the significance of variations in treatment means from morphological and physiological data analysis. Multiple means were compared using the Fisher's LSD test, and differences were considered significant at $p < 0.05$. Relative gene expression was quantified using the Delta-delta Ct formula as demonstrated in [19].

## 3. Results

### 3.1. Soil Water Content

The water content in the soil decreased from the highest of 43% to 13.7% as early as 3 days after water stress treatment, irrespective of the plant species (Figure 1). Thereafter, water continuously decreased at a slower rate until the final day (day 9) of moisture stress, when the lowest moisture content was recorded for Clm-02 plants (13.7%). After re-watering, the soil water content significantly increased for all genotypes to the highest of 25% at day 12.

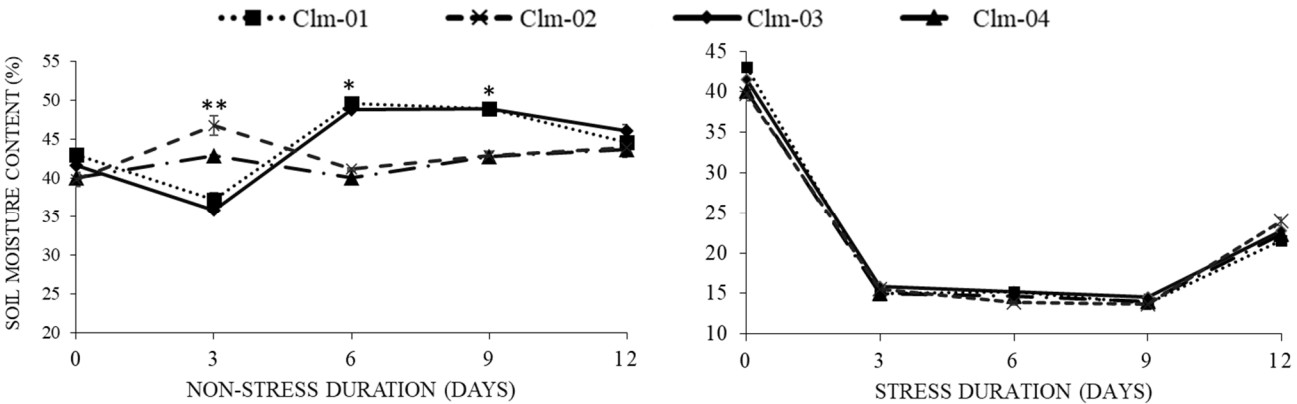

**Figure 1.** Soil moisture content response to drought stress in non-stressed (control) plants (**left panel**) and drought-stressed plants (**right panel**). Day 12 represents stressed plants after re-watering. Error bars show standard deviations of means for n = 3. Statistical analysis was performed for each day of data collection; * significant differences at $p < 0.05$, ** highly significant differences at $p < 0.05$, per ANOVA performed using R v.4.3.1 software (2023).

### 3.2. Physiological Parameters

The effect of water deficit on the physiological performance of watermelon cultivars is shown in Figure 2. All cultivars underwent a significant decline in stomatal conductance (gs), leaf relative water content (RWC), and chlorophyll content (SPAD) with decreased moisture. In addition, the cultivars differed significantly in their response to different stress levels. High RWC in wild watermelon was recorded for all stress days, with the highest value of 89 ± 0.82% recorded on day 3 (Figure 2B).

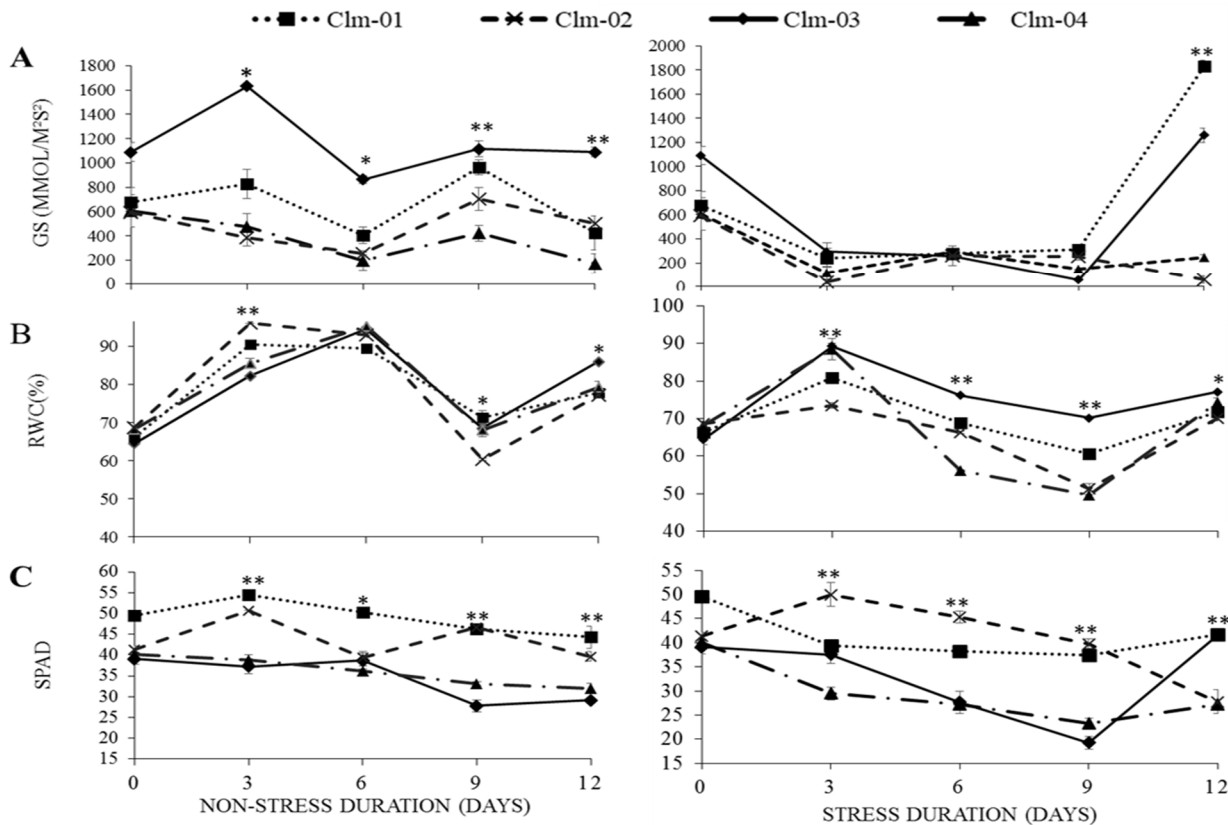

**Figure 2.** Effects of water stress on (**A**) stomatal conductance (GS), (**B**) relative water content (RWC), and (**C**) chlorophyll content, measured as SPAD, in non-stressed plants (control) (left panel) and drought-stressed plants (right panel). Day 12 represents stressed plants after re-watering. Error bars show standard deviations of means with n = 3. Statistical analysis was performed for each day of data collection; * significant differences at $p < 0.05$, ** highly significant differences at $p < 0.05$ per ANOVA performed using R v.4.3.1 software (2023).

A significant decline in stomatal conductance was observed as early as day 3 when the highest recorded value was $297.0 \pm 73.2$ in Clm-03. The lowest value on the same day was $35.5 \pm 14.4$ mmol/m$^2$s$^2$ in Clm-02 at. At day 9 of drought stress, the lowest recorded stomatal conductance was $55.1 \pm 6.3$ mmol/m$^2$s$^2$ in Clm-03. Under drought stress, the highest SPAD value was recorded for Clm-02 at $50 \pm 3$ on day 3 and the lowest value for Clm-03 at $19 \pm 1$ on day 9. In general, Clm-02 had the highest SPAD values in drought-stressed plants on all 3 stress days: $50 \pm 3$, $45 \pm 1$, and $40 \pm 1$, respectively, on days 3, 6, and 9. Clm-04 had the lowest SPAD values of $29 \pm 1.34$ and $27 \pm 0.37$ on days 3 and 6.

### 3.3. Chlorophyll Fluorescence (CF) Parameters

qL, Fv/Fm, and PS1 active centers:

The chlorophyll fluorescence parameters qL and PS1 active centers decreased slightly during the initial period of drought stress, while there was a significant decrease in Fv/Fm (Figure 3). Under drought stress, the lowest values of qL on all three stress days were recorded for Clm-01: $0.45 \pm 0.00$, $0.41 \pm 0.01$, and $0.50 \pm 0.01$ on day 3, 6, and 9, respectively, and the values on day 6 and 9 were significantly different from the other genotypes. The highest value of $0.78 \pm 0.02$ was recorded for Clm-04 on day 3. Overall, the highest average qL was recorded for Clm-04 in drought-stressed plants on all treatment days.

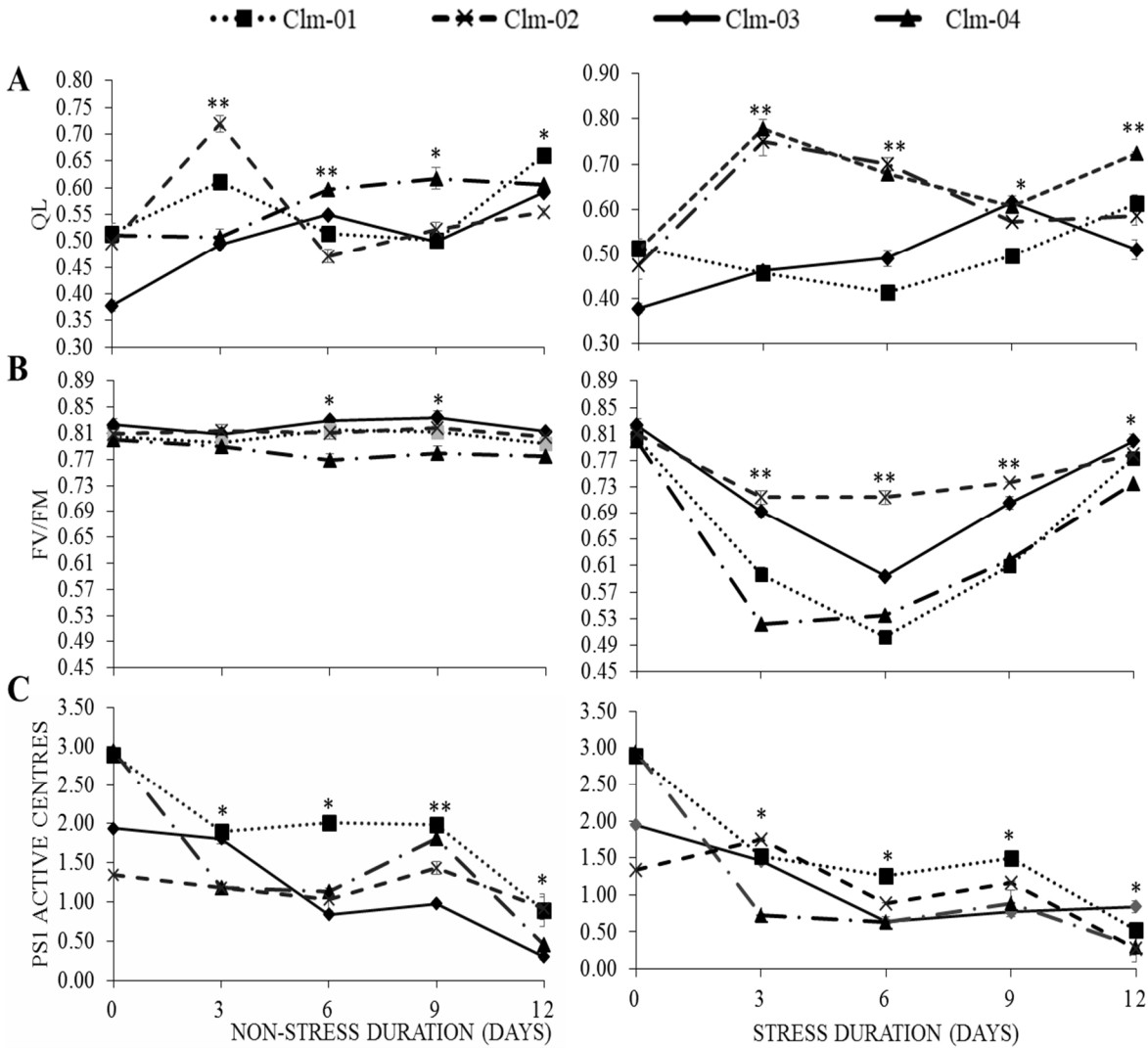

**Figure 3.** Effects of water stress on (**A**) photochemical quenching of variable chlorophyll fluorescence (qL), (**B**) quantum yield of photosystem II (Fv/Fm), and (**C**) photosystem I active centers in non-stressed plants (control) (left panel) and drought-stressed plants (right panel). Day 12 represents stressed plants after re-watering. Error bars show standard deviations of means for n = 3. Statistical analysis was performed for each day of data collection; * significant differences at $p < 0.05$, ** highly significant differences at $p < 0.05$ per ANOVA performed using R v.4.3.1 software (2023).

As shown in Figure 3B, there were no significant differences in the maximum quantum efficiency of PSII (Fv/Fm) values for non-stressed plants between either treatment days or genotypes throughout the study. Overall, a reduction in Fv/Fm was observed for all watermelon genotypes subjected to drought stress conditions. The trend of decline and recovery was similar for all genotypes but varied in magnitude. Clm-02 had significantly higher Fv/Fm values, which were maintained across all stress days, with a slight decrease from day 0, when it was $0.83 \pm 0.01$, and a decline thereafter to $0.71 \pm 0.01$, $0.71 \pm 0.01$, and $0.74 \pm 0.00$, respectively, on day 3, 6, and 9. The second genotype showing a slight decline was Clm-03, with values of $0.69 \pm 0.01$, $0.59 \pm 0.00$, and $0.71 \pm 0.01$, respectively, on the three stress days. The lowest value of $0.50 \pm 0.00$ was recorded for local watermelon on day 6.

Figure 3C shows that there was a less significant reduction in recorded PS1 AC values under drought stress. PSI active centers showed a fluctuating tendency, declining on days 3 and 6 and recovering on day 9, then decreasing again on day 12 for all watermelon genotypes under non-stress conditions. A similar pattern was observed under drought

stress treatment, with PSI active centers decreasing on days 3 and 6, increasing on day 9, and decreasing again on day 12.

PhiNPQ, PhiNO, and Phi2:

In general, when exposed to drought stress, PSII efficiency and the representative non-photochemical parameter (PhiNPQ) tended to increase, while PhiNO and Phi2 decreased with drought stress (Figure 4).

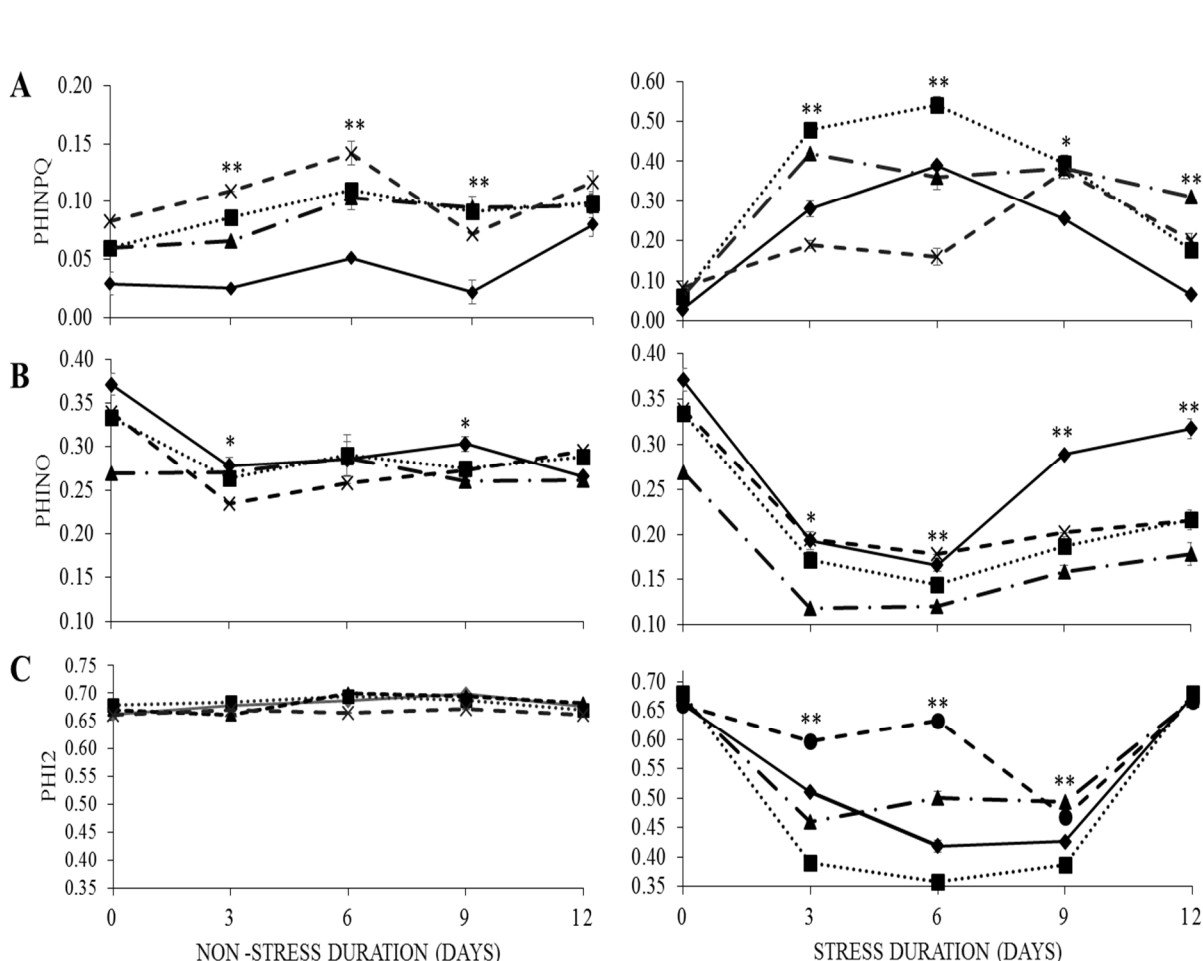

**Figure 4.** Effects of drought stress on (**A**) quantum yield of regulated energy dissipation in PS II (PhiNPQ), (**B**) quantum yield of non-regulated energy dissipation in PS II (PhiNO), and (**C**) effective quantum yield of photochemical energy conversion in PS II (Phi2) in non-stressed plants (control) (left panel) and drought-stressed plants (right panel). Day 12 represents stressed plants after re-watering. Error bars show standard deviations of means for n = 3. Statistical analysis was performed for each day of data collection; * significant differences at $p < 0.05$, ** highly significant differences at $p < 0.05$, per ANOVA performed using R v.4.3.1 software (2023).

Under drought stress treatment, Clm-01 had the highest PhiNPQ values on all three stress days, reaching $0.54 \pm 0.02$, the highest on day 6 (Figure 4A). Clm-02 had the lowest values on days 3 and 6 of stress, $0.19 \pm 0.01$ and $0.16 \pm 0.02$, respectively. On day 9, Clm-03 had the lowest value of $0.26 \pm 0.00$. Overall, under drought stress, PhiNPQ values for Clm-01 and Clm-03 gradually increased on day 3 and 6 of stress treatment, decreased on day 9 of stress, and continued to decrease on day 12 of stress, while for Clm-02 and Clm-04, the values increased on day 3, slightly decreased on day 6, then increased on day 9 of stress, and decreased on day 12.

### 3.4. Stomatal Morphology (Stomatal Aperture Measurements)

Overall, drought stress led to a reduction in stomatal aperture in all studied watermelon genotypes (Figures 5 and 6). Clm-02 had the largest average stomatal size compared to the other genotypes on all stress days, with the largest being 33.0 ± 1.3 μm on day 0. On day 9 of stress, Clm-03 had the smallest stomatal aperture size, 14.7 ± 0.7 μm, while Clm-02 had the largest, 23.1 ± 1.2 μm. On day 12 of re-watering, Clm-03 had a highly significant rapid recovery compared to the other genotypes, reaching 29.3 ± 2.5 μm, which was larger than on the other days. The recovery of Clm-01 and Clm-02 was delayed beyond day 12, with the stomatal size continuing to decrease.

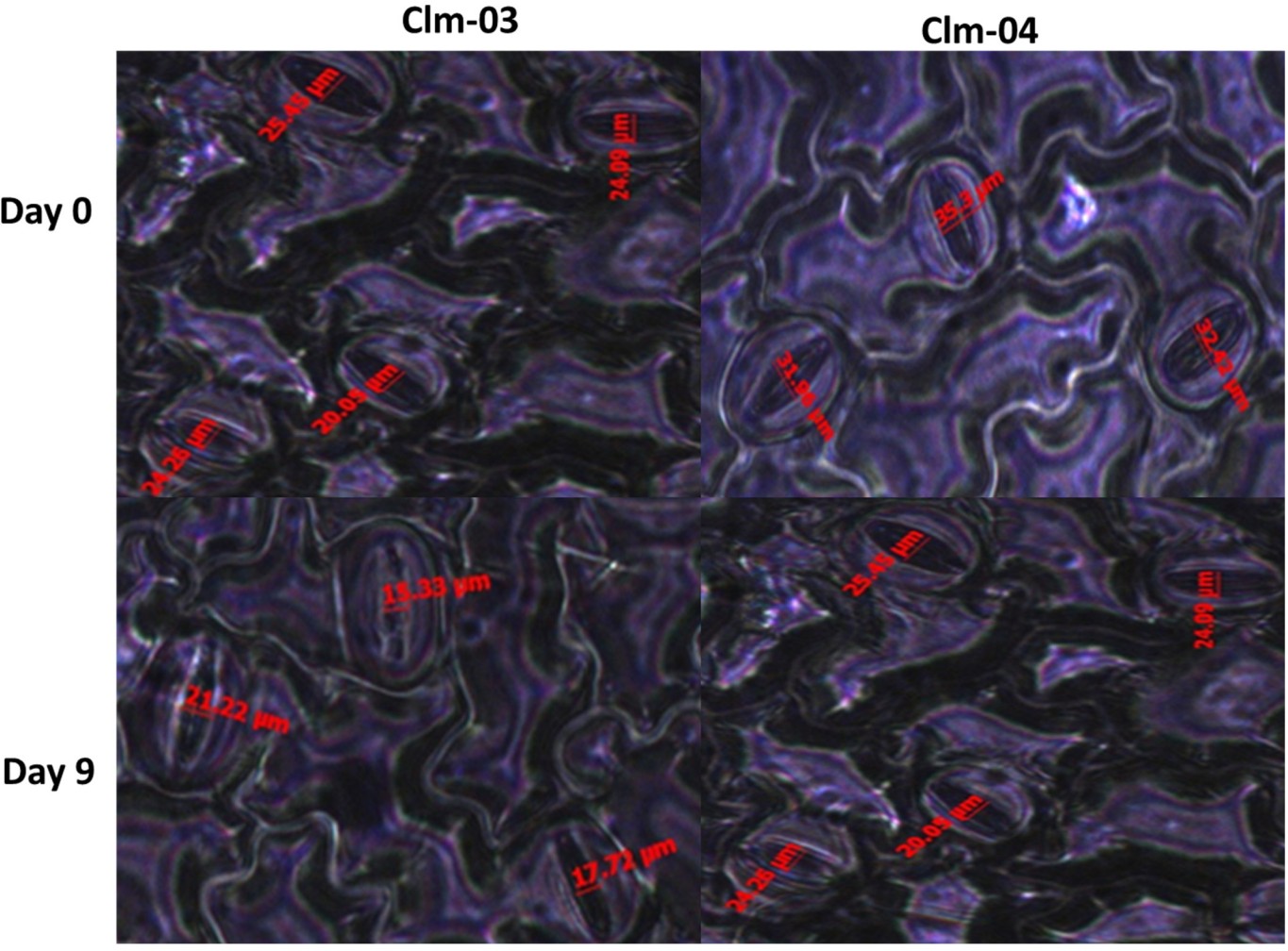

**Figure 5.** Representative images of the stomatal openings of two watermelon accessions under drought stress. Accessions shown here represent smaller stomatal openings (Clm-03) and larger stomatal openings (Clm-04) under drought stress in this study. Images were taken under an inverted biological microscope. Day 0 represents the first day before drought stress, and day 9 is the last day of drought stress.

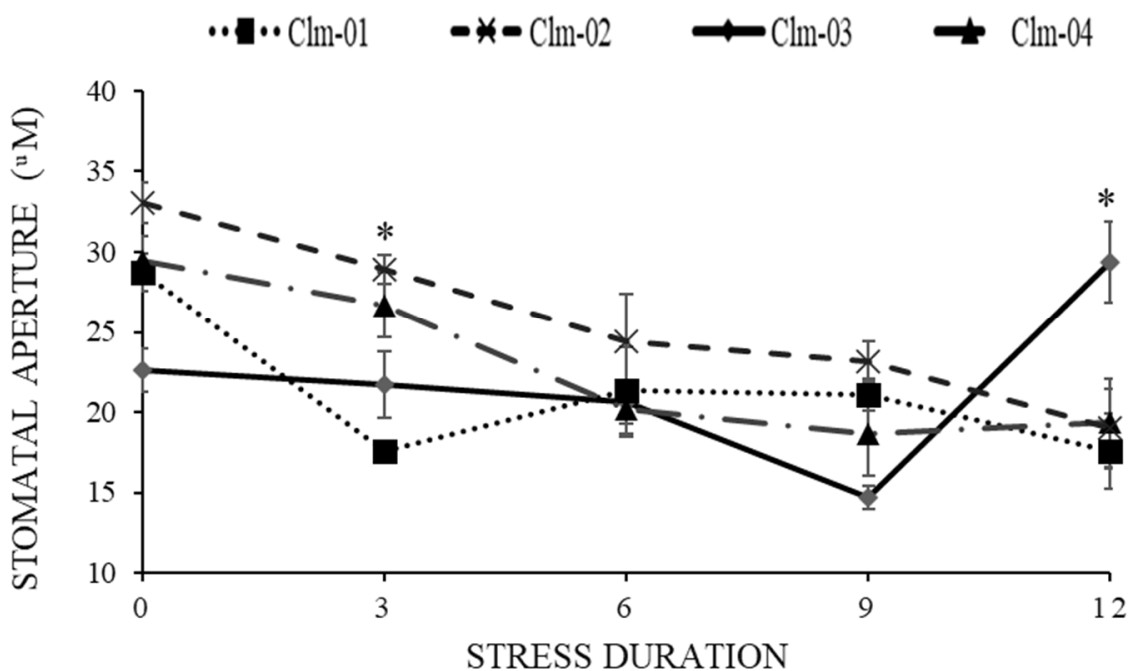

**Figure 6.** Effects of drought stress on the stomatal aperture response of four watermelon genotypes. Day 0 represents the first day before induced drought stress; days 3 to 9 represent the period of drought stress; and day 12 is the day after re-watering. Error bars show standard deviations of means for n = 5. Statistical analysis was performed for each day of data collection; * significant differences at $p < 0.05$, per ANOVA performed using R v.4.3.1 software (2023).

*3.5. Relative Expression of Plant Transcriptional Regulators with a Function in Stomatal Movement*

The expression patterns of genes related to stomatal aperture showed that the studied genes were related to stomatal movement (Figure 7). Clm-03 was found to have limited stomatal opening as drought stress progressed, as also shown by the expression of TFs *Cla016489* and *Cla004380* related to stomatal closure (Figure 7A,B). *Cla004380* in Clm-03 showed the highest increase on day 9 of stress, with about an eight-fold increase compared to the other genotypes. The other three accessions had the lowest expression of the studied TFs for stomatal closure, with their stomata remaining open even under extreme stress. The expression patterns of the TFs responsible for stomatal opening had contradictory results. *Cla017389* showed higher expression in Clm-01 and Clm-04 even under extreme drought stress, while Clm-03 and Clm-02 showed lower expression, suggesting that their opening was limited. The highest increase of *Cla017389* was observed in Clm-01 and Clm-04 on day 9 of drought stress, with five- and six-fold increases, respectively. The results for *cla017012* showed upregulation in Clm-02 and Clm-03 on day 6, then significant downregulation on day 9. Clm-01 and Clm-04 were not significantly different throughout the study, and no notable increase was observed as stress duration progressed.

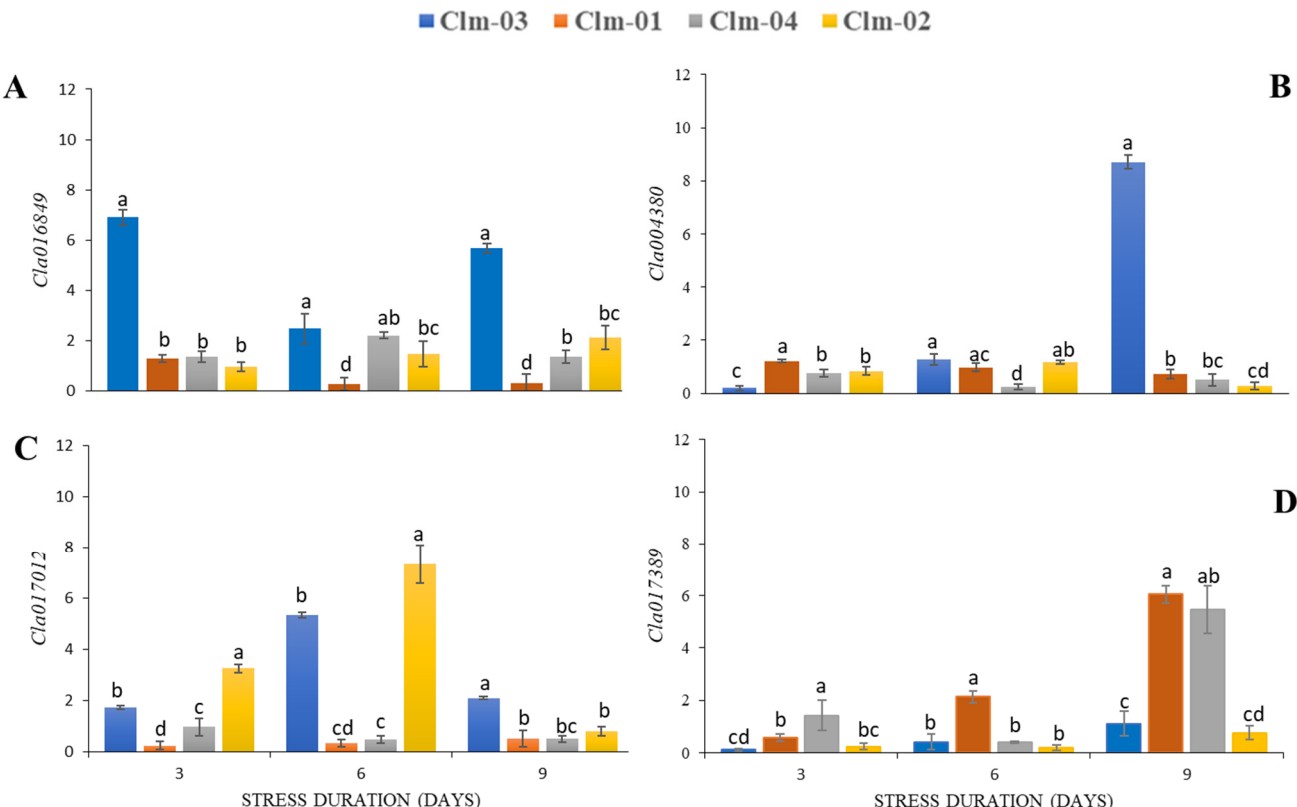

**Figure 7.** Results of gene expression analysis of four watermelon genotypes on three drought stress days (days 3, 6, and 9). Relative expression of the stomatal movement genes are shown as follows *Cla016849* (**A**) and *Cla004380* (**B**) stomatal opening genes and *Cla017012* (**C**) and *Cla017389* (**D**) stomatal closure. Delta-delta Ct relative quantification was used to estimate the mRNA levels of genes of interest (GOIs). Error bars show standard deviations of means for n = 5. Multiple means were compared using Fisher's LSD test at a 95% significance level, and significant differences among means are indicated by different letters above the bars.

## 4. Discussion

Root zone soil moisture is critical for crop growth; hence, soil moisture can serve as an indicator of drought-induced water stress in plants [20]. The optimal range of soil moisture content for crops depends on the specific plant species, and the range for most crops is between 20% and 60% [21]. In this experiment, drought stress resulted in a highly significant reduction in soil moisture in all studied watermelon genotypes, to about 13.7% in Clm-02 on day 9 of stress and the highest of 14.6% in Clm-03. This indicates that very high levels of moisture stress and decreasing soil moisture triggered drought stress in watermelon plants, with plant wilting observed as early as day 3 of drought stress. Root zone moisture has been found to be directly related to stomatal movement through root-to-shoot signals [22]. Root-to-shot signals vary from plant to plant and are a critical component of the response to drought.

Stomatal closure has been noted as one of the first plant responses to drought; it is crucial in determining overall plant survival under moisture stress [23]. It is known to be a critical mechanism that plays an important role in reducing water loss and increasing $CO_2$ diffusion resistance in the mesophyll; thus, crops that actively regulate their stomata during stress have a greater chance of survival. In this study, a decrease in Gs was recorded for all watermelon genotypes under drought stress. The decrease was larger in Clm-03 than in other genotypes under prolonged stress (Figure 2A), which suggests that Clm-03 has better mechanisms to adapt to moisture stress conditions.

Relative water content, a crucial indicator of the degree of plant hydration, is important for optimum physiological status and growth [24,25]. In the current study, RWC decreased

in all watermelon genotypes with increasing water stress, and wild watermelon had higher RWC on all stress days (Figure 2B). The high RWC level in Clm-03 leaves indicates greater drought stress tolerance, as expected given the natural habitat of this crop. Our findings are consistent with the results of experiments with wild watermelons exposed to a water deficit [26]. Additionally, several studies on melon (*Cucumis melo* L.) and watermelon (*Citrullus lanatus*) species demonstrated parallel results for the effects of drought on relative water content in plants [27–29]. This suggests that drought-tolerant plant species can maintain high leaf-relative water content under drought-stress conditions. Stomata have been noted to play an important role in the hydrological cycle, which is responsible for regulating the transpiration of water from plants to the atmosphere, and plants with limited stomatal movement early in drought stress have reduced moisture loss, which improves their tolerance to moisture stress [30]. The results show that stomatal opening plays a significant role in the maintenance of RWC in tolerant plants, with wild watermelons having fewer changes in RWC as drought stress progresses. This suggests that stomatal closure is important to this ability and helps wild watermelons tolerate drought stress to some extent.

Chlorophyll content (SPAD) is a crucial indicator to monitor plant performance under abiotic stress. Drought stress has been noted to accelerate chlorophyll decomposition in various plants [29]. As expected, the results of this study show a negative impact of drought stress on Chl content, with a significant difference among genotypes (Figure 2C). Similarly, another study [31] reported that SPAD values significantly decreased from 35.5 to 22.4 in soybean (*Glycine max* (L.) Merr.) under drought stress. In the present study, SPAD values decreased with increasing water stress duration; however, Clm-02 had significantly higher SPAD values on all stress days, with the highest of 50 on day 3, while the lowest (19) was recorded in Clm-03 on day 9. This result suggests that Clm-02 has a mechanism that protects chlorophyll from degradation under drought stress. Interestingly, Clm-03, which is thought to be more tolerant, had lower chlorophyll values during stress treatment, but upon re-watering, its recovery was better than the other genotypes, as shown in Figure 2C. However, it is possible that quick loss of chlorophyll under drought stress may be an adaptive protective response to reduce photosynthesis, which can be destructive when it takes place under drought stress [32]. A decrease in total chlorophyll under drought stress suggests a reduced capacity for light harvesting. Since ROS production is mainly driven by excess energy absorption in the photosynthetic apparatus, when the balance between light absorption and carbon fixation is disrupted due to drought stress, this could be avoided by the degradation of absorbing pigments [33].

Fv/Fm, PhiNPQ, qL, and Phi2 are the most important Chl fluorescence parameters and are broadly used in plant stress response studies [34–37]. An inverse relationship between qL and PhiNPQ and between Phi2 and PhiNPQ has been noted [36]. In this study, Phi2 decreased with increasing PhiNPQ, and values were in line with previously reported results [36,38].

PhiNPQ, which is the amount of dissipated excessive irradiation, serves a protective function in plants under drought stress [39]. As expected, PhiNPQ values increased with increased drought stress duration. The results in Figure 4A show that under drought stress treatment, local watermelon had the highest PhiNPQ values across all three stress days, with the highest on day 6: $0.48 \pm 0.00$, $0.54 \pm 0.02$, and $0.40 \pm 0.02$ for days 3, 6, and 9, respectively. Clm-02 had the lowest values on days 3 and 6 of stress, $0.19 \pm 0.01$ and $0.16 \pm 0.02$, respectively. Drought stress makes the electron transfer chain saturated and increases proton accumulation, thus increasing NPQ [40]. Higher NPQ values indicate the ability to mitigate the negative effects of drought stress at the chloroplast level, as these organelles can dissipate the excess excitation energy [41]. PhiNPQ was found to be significantly high in a study on tolerant tomatoes (*Solanum lycopersicum*) when exposed to drought stress [42]. The results indicated that it was unlikely that significant photo-inhibition occurred, suggesting enhanced protection of the photosynthetic apparatus. This

phenomenon has also been related to an observed increase in antioxidant levels, especially carotenoids, in the chloroplast [43].

Fv/Fm, the ratio of variable fluorescence to maximal fluorescence, measures the maximum efficiency of PSII (if all PSII centers are open) [35]. A decrease in Fv/Fm reflects a reduction in light use efficiency in plants [44,45]. An increase in Fv/Fm is correlated with reduced energy loss as heat [46–48]. In this experiment, it was observed that under drought stress, Fv/Fm decreased with increased PhiNPQ (Figures 3A and 4A). In this experiment, Fv/Fm was the lowest at 0.50 for Clm-01 on day 6. The highest values were recorded in Clm-02 across all three stress days: 0.71, 0.71, and 0.74, respectively, which represents only a 0.12% reduction compared to the average Fv/Fm values in control plants. This may indicate that Clm-02 has a very important mechanism by which it is able to reduce damage to the photosystem and continue the photosynthetic process even under moisture stress. This may be true, in that Clm-02 usually produces large fruits even during extreme drought seasons [49].

Induced water stress caused a less significant reduction in photosystem I active centers in this experiment. In contrast, in another study, the proportion of PSI-active centers in three maize hybrids was higher under drought stress [50]. This supports the idea that the additional PSI active centers have a role in cyclic electron transport (CEF) to stimulate the proton-motive force that results in non-photochemical quenching (NPQ) to protect PS II [51]. CEF also helps in avoiding the over-reduction of PSI and the generation of ROS under drought stress. In our study, the highest PS1 active center values were recorded for Clm-01 on days 6 and 9 of intensified drought stress, and this may indicate the ability to protect PS II under drought stress.

As the growth and survival of plants in drought-stress conditions are related to water availability, stomata play an important role in managing water loss through transpiration [52], thus aiding plant growth and survival. Minimal stomatal opening is an expected response in plants subject to drought stress [53]. Drought stress resulted in reduced stomatal aperture in all watermelon genotypes in this study (Figures 5 and 6), and a highly significant reduction was recorded under extreme drought stress (days 6 and 9). The lowest value for stomatal aperture (14.7) was recorded in Clm-03 on day 9 of drought stress, and this is in line with the drought stress avoidance mechanism reported in [54,55]. Another study [54] reported that both medium and severe droughts led to significant decreases in stomatal aperture in drought-stressed maize (*Zea mays*) plants compared with well-watered plants. It was suggested that by regulating the stomatal pore aperture, plants can optimize their $CO_2$ uptake for photosynthesis while minimizing water loss, which improves water use efficiency and ultimately drought stress tolerance [56]. The stomatal movement has been shown to play an important role under different environmental conditions, and highly responsive stomata might achieve an important response to abiotic stress [56]. In the current study, the two probable tolerant landraces showed rapid stomatal responses, with restricted stomatal opening, thus limiting moisture loss and aiding stress tolerance. Of all the accessions, the hybrid watermelon, which had higher stomatal aperture movement during drought stress, showed signs of susceptibility to drought stress, indicating that the timely movement of stomata is important in the tolerance mechanism of plants.

Transcription factors, which naturally act as master regulators of cellular processes, are suspected to be involved in modifying complex traits, such as responses to environmental stresses [57]. The phytohormone abscisic acid (ABA) is an important regulator of plant responses to abiotic stress [58]. Drought stress results in the accumulation of ABA, which initiates many adaptive responses [59], including plant transcriptional regulators. Under water-deficit conditions, upregulated genes participate in the ABA signaling pathways, such as ABA-induced stomatal closure, which reduces transpirational water loss from plants [60,61].

Transcription factors *Cla016849* and *Cla004380*, which have been documented to have active roles in guard cell ABA signaling and stomatal closure [62], were studied. The highest expression of both genes was recorded in wild watermelon (Figure 7A,B) on day

9 of stress, and this result correlates with the morphological expression of the stomatal aperture (Figures 5 and 6). This has been suggested to be the most critical mechanism used by wild watermelons to manage the internal photosynthetic apparatus and maintain internal moisture, thus avoiding the adverse effects of drought stress [55,63]. In another study [62], overexpression of Arabidopsis with the cyclophilin ROC3 gene led to reduced stomatal closure in mutants compared to the wild-type, showing the important role of ROC3 as a homolog in regulating stomatal closure.

*Cla017012* and *Cla017389* (Figure 7C,D), transcription factors involved in abscisic acid-mediated stomatal opening, were also studied. The results for *Cla016849* were not conclusive and thus are not discussed in this paper. *Cla017389*, which is involved in stomatal opening and is ABA-dependent [64], had significantly high expression in Clm-01 and Clm-04 on day 9 of drought stress. It had the lowest expression in Clm-03 on days 3 and 9. These results suggest that the two genotypes' stomata remain open even under extreme drought stress, which may be associated with their susceptibility, as continued photosynthesis during stress leads to damage to the photosynthetic apparatus, in turn leading to adverse effects on the plant. A study of yellow horn (*Xanthoceras sorbifolium*) revealed that stomatal opening under drought stress could result in limited carbon dioxide diffusion in leaf photosynthesis as well as unnecessary water loss when the stomata remain open after photosynthesis has reached saturation [65]. The *Cla017012* gene family (MYB60) has been suggested to be negative modulators of jasmonate, which are responsible for stomatal closure in plants [66], and this was further confirmed in a study showing slower stomatal opening in Arabidopsis MYB60 mutants [67]. Our results show that *Cla017012* (MYB60) was upregulated in the two accessions on day 6 and downregulated thereafter, suggesting a reduced negative modulation of jasmonate, thus allowing reduced stomatal opening as a tolerance mechanism to reduce moisture loss.

Genetic factors have been shown to play an important role in stomatal movement, as shown by several studies in which overexpression of genes [68–70] related to stomatal movement resulted in significant changes that helped plants to withstand abiotic stress like drought. Thus, genes from tolerant species can play a significant role in improving susceptible relatives or even other species. Transcription factors related to stomatal closure were highly expressed in the two probable tolerant accessions in this study and can be used for improving other crop species.

### 5. Conclusions

The four watermelon (*Citrullus lanatus*) accessions evaluated in this study represent the major cultivated species of watermelon in Botswana and most of southern Africa. These are normally cultivated annually under varying climatic conditions and have been shown to have varying levels of drought tolerance, as observed in most traditional farmers' fields. The results of this study further confirm the observations that Clm-03 and Clm-02 are more tolerant than Clm-04 and Clm-01. This is usually the case in traditional fields, where even during the worst drought seasons, there are always Clm-02 and Clm-03 plants and fruits available. This highlights the two landraces as important research subjects to fully understand their tolerance mechanisms, which can be harnessed to improve susceptible crop species. Importantly, the stomatal responses, as observed in morphological and molecular evaluations, indicate that stomatal movement is a significant component of species tolerance. The tolerant landraces showed rapid closure when exposed to stress, and the response was a bit delayed in susceptible accessions; interestingly, the reverse was also true for the studied accessions. This highlights an important trait that needs to be fully studied and improved in order to successfully develop climate-smart crops that can survive in extreme drought conditions, thus contributing to alleviating the dwindling of food security.

**Author Contributions:** Conceptualization, K.M. and G.M.; methodology, K.M.; validation, G.M.; formal analysis, K.M.; investigation, K.M., L.T.S. and M.N.N.; resources, G.M.; writing—original draft preparation, K.M.; writing—review and editing, K.M., G.M. and U.B.; supervision, G.M. and

U.B.; project administration, G.M.; funding acquisition, G.M. All authors have read and agreed to the published version of the manuscript.

**Funding:** This research was funded by an SG-NAPI award supported by the German Ministry of Education and Research, through UNESCO-TWAS (4500454040) and the Botswana University of Agriculture and Natural Resources.

**Data Availability Statement:** Data are contained within the article.

**Conflicts of Interest:** The authors declare no conflict of interest.

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
