# Peer review of "Morphological, Physiological, and Molecular Stomatal Responses in Local Watermelon Landraces as Drought Tolerance Mechanisms"

_horticulturae, doi:10.3390/horticulturae10020123_

Round 1

Reviewer 1 Report

Comments and Suggestions for Authors

The main weakness of the manuscript concerns the lack of statistical analysis: the author claim to have performed ANOVA but results are not shown. A large part of the discussion and conclusions seems not supported by the data shown.

Specific comments are reported in the attached file

Author Response

We would like to take this opportunity to thank the reviewer for taking their valuable time to review our manuscript and make comments that will help in improving the quality of our manuscript.

The comments made were very vital and we have addressed them in the manuscript and highlighted the revised areas. We have also provided a point-by-point response to all the editors and reviewers comments.

Reviewer 2 Report

Comments and Suggestions for Authors

Comments for the Author:

In the manuscript of “Morphological, Physiological And Molecular Stomatal Responses of local watermelon landraces As Drought Tolerance Mechanisms”. This study has provided an important insight into stomatal responses of probable tolerant watermelon accessions and suggest an improvement in the stomatal aperture on susceptible domesticated species will also improve the tolerance level. In my opinion, just a few changes are required.

1.      In table 1, The first letter of the word ‘landrace’ should be capitalized. Why did the author choose three landraces and only one hybrid material? Please provide appropriate reasons.

2.      The lines in Figures 1 to 4 cannot distinguish the differences among the four watermelon materials well.

3.      In Figure 5, the numerical values labeled for the size of the pores are not clear, and the brightness of the photo is inconsistent. Please provide a clearer photo.

4.      In Figure 7, the number format of the horizontal and vertical coordinates is inconsistent. Please maintain consistency.

5.      In Figure 7, the expression patterns of the four transcription factors selected by the author are not the same. How to explain this.

6.      The manuscript was not well written with many typos, grammar errors, redundancies, and inaccurate statements. A major revision in both science and language is needed to improve readability and clarity.

Comments on the Quality of English Language

The manuscript was not well written with many typos, grammar errors, redundancies, and inaccurate statements. A major revision in both science and language is needed to improve readability and clarity.

Author Response

(The authors gave the same response as above.)

Round 2

Reviewer 2 Report

Comments and Suggestions for Authors

I think this manuscript has been revised and can be accepted

Author Response

Comment 1. Review grammar

Response: The Manuscript has been given an extensive revision and editing through the MDPI editing services and major revisions on grammar were done and highlighted throughout the manuscript by track changes

Comment 2. It would be desirable to introduce here the code of the genotypes since in the next sentence are mentioned.

Response: We thank the reviewer for the thoughtful suggestion which has been incorporated to included codes of genotypes

Comment 3. It would be advisable to be more accurate, which accessions?

Response: The objective has been rephrased to be more accurate by include the accessions code that are a primary focus

Comment 4. It makes no sense to multiply and divide by the same number.

I knew this other formula:

WC = [(FW − DW)/FW] × 100

Could it be some kind of typo.

Response: We thank the reviewer for highlighting the typing error. The formula has been corrected by inserting the omitted minus (-) sign

Comment 5. It would be desirable to maintain the spelling: below it is written with hifon and CT out of the brackets

Response: A similar spelling has been maintained throughout the manuscript.

Comment 6. It makes no sense, it would be advisable to rewrite this caption.

Response: The caption has been revised by both the authors of the manuscript and the MDPI editing service.

Comment 7. non-stressed"

In the same manner as the caption of figure 2, it make no sense to speak about effect of stress in non-stressed plants. A suggestion is to speak about treatment duration and specify which figure correspond to water stress treatment or control.

Response: The caption has been revised by both the authors of the manuscript and the MDPI editing service, and this has also been extended to all other figures

Comment 8. There is no day 15 in the figure.

Response: The caption has been revised and corrected to delete the day 15

Comment 9:

From reviewer 1:

"what about stomatal conductance, (fig. 2), that seems very similar in all varieties under stress? and about Cla017012 that is highly expressed in Cml3? what about ANOVA results in figure 6, where differences seems very poor?"

Response: The results obtained in the study under stress showed minimal differences in stomatal conductance between the accession studied thus the graph showing near similar stomatal conductance’s in all varieties. The differences were not significant.

The high expression of Cla017012 in Clm-03 has been addressed in the results and also in the discussion, we believe the reviewer might have missed it.

The ANOVA results in figure 6 only shows significant differences on day 3 and 12 only and the other days there was no significant differences. The caption has also been revised to explain what the figure entail

Comment 10. From reviewer 1: "not true for qL"

Response: The sentence has been revised